# Meeting the 2030 END TB goals in the wake of COVID-19: A modelling study of countries in the USAID TB portfolio

Nimalan Arinaminpathy[1]*, Ya Diul Mukadi[2], Amy Bloom[2], Cheri Vincent[2], Sevim Ahmedov[2]

1 MRC Centre for Global Infectious Disease Analysis, Imperial College, London, United Kingdom, 2 United States Agency for International Development, Washington, DC, United States of America

* nim.pathy@imperial.ac.uk

## Abstract

Progress towards the 2030 End TB goals has seen severe setbacks due to disruptions arising from the COVID-19 pandemic. For governments and international partner organizations supporting the global TB response, there is a need to assess what level of effort is now needed to reach these goals. Using mathematical modelling, we addressed this question for the countries being supported by the United States Agency for International Development (USAID). We aggregated the 24 countries in the USAID portfolio into three geographical country groups: South Asia; sub-Saharan Africa; and Central Asian Republics/Europe (CAR/EU). From 2023 onwards we modelled a combination of interventions acting at different stages of the care cascade, including improved diagnostics; reducing the patient care seeking delay; and the rollout of a disease-preventing vaccine from 2025 onwards. We found that in all three country groups, meeting the End TB goals by 2030 will require a combination of interventions acting at stages of the TB care cascade. Specific priorities may depend on country settings, for example with public-private mix playing an important role in countries in South Asia and elsewhere. When a vaccine becomes available, its required coverage to meet the 2030 goals will vary by setting, depending on the amount of preventive therapy that has already been implemented. Monitoring the number-needed-to-test to identify 1 person with TB in community settings can provide a useful measure of progress towards the End TB goals.

## Introduction

In relation to tuberculosis (TB), the Sustainable Development Goals (SDG) for 2030 call for an 80% reduction in annual incidence rates and a 90% reduction in TB mortality, by 2030 relative to 2015 [1]. In recognition of the importance of TB as a global health problem, a United Nations High-Level Meeting (UN HLM) was held in 2018 [2]. This meeting was followed by targets being set, for numbers of individuals with TB who would be diagnosed and treated, and for numbers of at-risk individuals receiving tuberculosis preventive therapy, between 2018 and 2022. Although recent years have seen important improvements in TB services in many

**Data Availability Statement:** All relevant data are within the paper and its Supporting Information files.

**Funding:** This work was supported by USAID (through a contract to NA). YDM, AB, CV and SA are employees of USAID, and their respective contributions are specified in the author information. The funders otherwise had no role in the study.

**Competing interests:** The authors have declared that no competing interests exist.

different settings [3], much more needs to be done in order to accelerate the slow declines in TB burden that were occurring in the years leading up to 2019.

Mathematical modelling has been a helpful tool in anticipating the increase in programmatic effort and funding that needs to occur, in order to meet the End TB goals. For example, early modelling showed the need for a comprehensive set of interventions, combining all available measures, as well as new tools for diagnosis, treatment and prevention [4, 5]. This message was echoed in more recent modelling analysis tailored to countries in the WHO South-East Asian Region, that also highlighted the importance of coordinating all TB services, including in the private sector that plays such an important role in the management of TB in the Region [6, 7]. Moreover, modelling analysis was key in informing the targets that were set during the 2018 UN HLM [8].

However, the COVID-19 pandemic–and the disruptions to healthcare services that followed–had severe adverse effects on the global TB response. During lockdowns and other movement restrictions, symptomatic patients faced heightened barriers to seeking care, while crucial healthcare resources, including personnel and diagnostics, were diverted to the COVID response [9]. As a result, TB case detection fell sharply, with global notifications reducing by over 20% in 2020 [10]. Such developments are likely to have substantially increased TB incidence and mortality, because of increased opportunities for transmission arising from an expanded pool of undetected, untreated TB [3, 11].

It is now important for international partners, health agencies and national TB programmes to assess how the SDG 2030 goals should be approached, in the wake of COVID-related disruptions. For example, what scale of interventions are needed to meet the SDG 2030 goals? Will meeting the 2018 UN HLM targets still signify progress towards these goals? What alternative approaches might also now be used to monitor progress? Using mathematical modelling, we sought to address these questions for 24 countries whose TB programmes are currently receiving bilateral support from USAID, the international development agency of the USA.

## Methods

### Country groups

We first grouped the 24 countries in the USAID TB portfolio into three different geographies: South Asia, sub-Saharan Africa, and Central Asia/Eastern Europe (CAR/EU) (see Table 1). Fig 1 illustrates the distinct TB epidemiology associated with each of these groups. HIV is a major driver of TB incidence in sub-Saharan Africa, while rifampicin-resistant TB accounts for a disproportionate amount of TB burden in CAR/EU. Finally, for most countries in the

**Table 1. Groupings of countries in the USAID portfolio into the different country groups.**

| South Asia | Sub-Saharan Africa | Central Asia and Europe (CAR/EU) |
|---|---|---|
| Afghanistan | DR Congo | Kyrgyz Republic |
| Bangladesh | Ethiopia | Tajikistan |
| Burma | Kenya | Ukraine |
| Cambodia | Malawi | Uzbekistan |
| India | Mozambique | |
| Indonesia | Nigeria | |
| Pakistan | South Africa | |
| Philippines | Tanzania | |
| Viet Nam | Uganda | |
| | Zambia | |
| | Zimbabwe | |

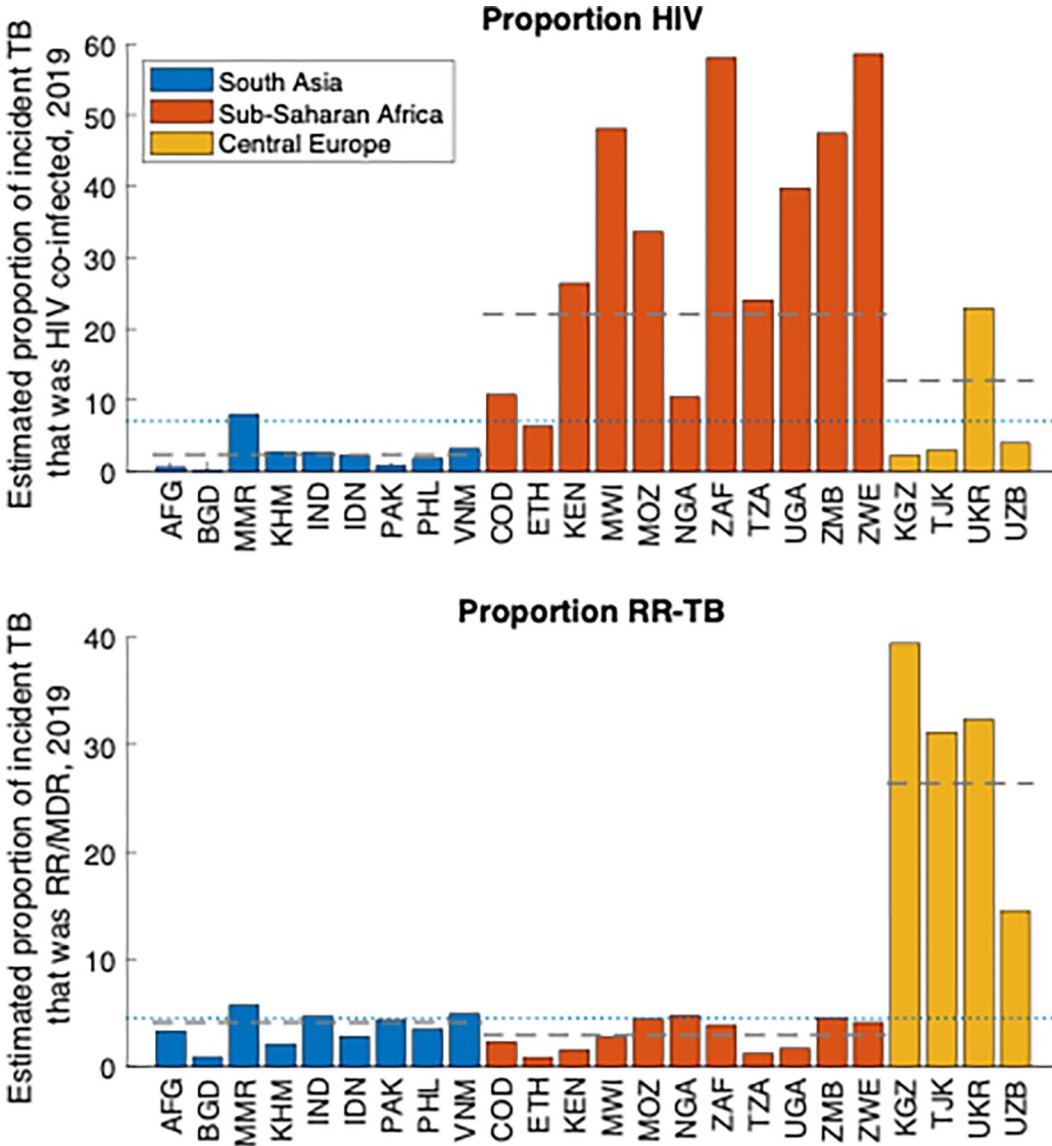

**Fig 1.** Comparison of different countries in the USAID portfolio, according to the proportion HIV/TB coinfection (top panel) and the proportion of incident TB that is rifampicin-resistant (bottom panel). Colours denote the different geographic groupings listed in Table 1. For each country group, horizontal dashed lines show average values, when weighted by country population size. The horizontal dotted line shows the population-weighted average over all countries in the portfolio.

South Asia country group, the private healthcare sector plays an important role in the management of TB, and needs to be engaged with effectively, to fully coordinate the TB response.

Accordingly, we developed three different model structures, one for each country group: for South Asia we developed a model taking account of public and private sectors, and capturing the lower standard of TB care (in particular, the less favourable diagnostic and treatment outcomes) that is generally observed in the private sector [12–14]. For sub-Saharan Africa, we modelled the role of HIV in driving the TB epidemic. We took account of trends in HIV incidence; HIV/TB coinfection; and the expansion of ART coverage. For CAR/EU we incorporated the acquisition and transmission of rifampicin-resistant TB, taking into account outcomes of second-line treatment, as well as current levels of drug sensitivity testing at the

point of TB diagnosis. For simplicity we did not aim to model each country individually, and instead modelled interventions at the level of the country groups shown in Table 1.

## Calibration

We performed the calibration in two steps: (i) calibrating to 2019 (i.e., pre-COVID data), and then (ii) to capture the effects of COVID-related disruptions, matching to quarterly notifications reported to WHO from 2020 onwards. In the first step, within each country group we aggregated country data in a population- weighted way, to give the indicators shown in Table 2. We performed calibration using Bayesian methods, as follows: We constructed likelihood terms by modelling each of the indicators shown in Table 2 using a log-normal distribution, and then taking a product over all likelihood terms for a given country group. We assumed uniform priors for each of the model parameters shown in Table A in S1 Text. With the posterior density thus constructed, we sampled from this density using adaptive Bayesian MCMC [15]. After discarding the 'burn-in' and thinning, we drew 250 samples from the posterior density for subsequent analysis.

In the second step, we used the posterior distribution sampled in the first step, to simulate the model to the end of 2019. From January 2020 onwards, we took simulated treatment initiations as a proxy for notifications. Since 2020, countries have been reporting notification data on a monthly or quarterly basis, to WHO. This data provides valuable information for the extent to which TB services were affected by the COVID pandemic, and form the basis of current WHO estimates of TB burden [3]. We followed the same approach, assuming that reductions in notifications—compared to 2019 levels—are attributable to delays in diagnosis and treatment initiation. Working with quarterly notification data, we aggregated country data to the level of country groups. For each country group, we then adjusted the rate-of-diagnosis on a quarterly basis, in order for modelled treatment initiations to match the notification timeseries as closely as possible. Because notification data is not available stratified by HIV or drug resistance status, nor by public or private sectors, we assumed for simplicity that disruptions to diagnosis were the same across all these strata.

## Interventions

In light of work prior to COVID disruptions showing the need for comprehensive interventions to meet the End TB goals [16, 17], we modelled the interventions listed in Table 3.

**Table 2. Pre-COVID calibration data used for each of the country groups.** All data, including uncertainty intervals, were drawn from the WHO global TB database.

| Country group | Indicator | | Calibration target |
|---|---|---|---|
| **South Asia** | Incidence per 100k population, 2019 | | 235 [166–317] |
| | Mortality per 100k population, 2019 | | 30 [26–34] |
| | Notifications per 100k population, 2019 | | 187 [168–206] |
| **Sub-Saharan Africa** | Incidence per 100k population, 2019 | All | 263 [169–397] |
| | | HIV +ve | 71.2 [45.9–102] |
| | Mortality per 100k population, 2019 | HIV -ve | 39.7 [24.2–59.7] |
| | | HIV +ve | 19.1 [10.5–30.7] |
| | ART coverage, 2019 (percent) | | 65.6 [54.2–80.1] |
| | HIV prevalence, 2019 (percent) | | 2.9 [2.6–3.4] |
| **Central Asia/Europe** | Incidence per 100k population, 2019 | All | 73.4 [52.9–103] |
| | | RR-TB | 20.9 [14.5–28.8] |
| | Notifications per 100k population, 2019 | All | 65.1 [58.6–71.6] |
| | | RR-TB | 12.0 [10.8–13.2] |

**Table 3. Interventions modelled for each of the country groups.** As noted in the text, these interventions mirror those modelled in the recent Global Plan to End TB [20]. Assumed levels of coverage are only illustrative, and do not necessarily represent the only possible scenarios for meeting the 2030 End TB goals. With the exception of vaccination, we assumed that all interventions would be initiated in 2022 and scaled up in a linear way over the next three years, to reach the coverage levels shown. For the TB vaccine scenario, we modelled a scenario where a vaccine is licensed by 2025, and scaled up over the subsequent three years.

| Intervention | Country groups applied to | Description |
|---|---|---|
| Public-private mix | South Asia only | Engage with 90% of private providers, improving diagnostic and treatment outcomes to same levels as in public sector |
| Improved second-line treatment outcome | CAR/EU only | Second-line treatment success improved to 85% |
| Improved diagnostics, routine TB services | All country groups | Modernise TB diagnostics throughout routine TB services (public and private) so that 90% of people with symptomatic TB are diagnosed per careseeking attempt |
| Upstream case-finding | All country groups | Among those with symptomatic TB, active case-finding and demand generation to decrease the delay-to-diagnosis by 30% |
| TB preventive therapy, risk groups | All country groups | Per WHO guidelines, full uptake of TPT amongst all-age household contacts and PLHIV (assuming 60% efficacy of TPT). |
| TB vaccine | All country groups | Post-exposure vaccine with 60% efficacy and providing immunity for 10 years, rolled out to achieve a given proportion of people with TB infection having vaccine-induced immunity. This proportion is determined for each country group in order to meet the 2030 goals, as follows: South Asia 72%, Sub-Saharan Africa 40%, CAR/EU 72%. |

Further information on these interventions is provided in the supporting information. The levels of intervention coverage shown below are purely illustrative scenarios, to show the level of effort that may be required to meet the 2030 goals.

We assumed all interventions would be rolled out starting in 2022 and scaled up linearly to achieve full coverage in three years, with the exception of vaccination, which was assumed to be rolled out starting in 2025. We simulated the impact of all interventions listed in Table 3, when acting in successive combination. We also examined whether it would still be possible to meet the 2018 UN HLM notification targets in each of the country groups, as a result of measures to increase TB case detection: that is, with all interventions short of preventive therapy and vaccination. The 2018 targets related to cumulative case-finding between 2018 and 2022, but progress towards these targets was interrupted during the COVID pandemic. We thus examined whether they could be attained in the period between 2022 and 2026.

Finally, as an alternative to notification targets, we simulated the potential role of the number-needed-to-test (NNT) to identify 1 individual with TB, as a measure of progress towards the End TB goals. Borrowing from ongoing work in India [18], this approach is based on the intuition that, as TB burden declines in any given setting, it should become necessary to test more and more individuals in order to find a single person with TB: that is, that NNT should increase over time. We considered three different types of NNT:

- *Amongst symptomatic individuals already being tested in routine TB services.* An advantage of this approach is that it can be readily monitored with existing systems. However, because this approach only involves those who have come forward for care, it may not be representative of the overall burden of TB in the community. For simplicity, we assumed that the NNT was 10 in 2019, prior to the pandemic, broadly consistent with programmatic data (see e.g. ref [19]).

- *Amongst symptomatic individuals being screened in active case-finding.* This approach has the advantage of potentially giving a more accurate measure of TB burden in the community,

than focusing only on symptomatics presenting for care. Although it relies on the implementation of active case-finding, modelling results suggest that these activities are required to meet the End TB goals [20]: assuming that active case-finding involves symptom screening, we model a scenario where the NNT is monitored, amongst symptomatic individuals being tested. For simplicity, we assumed that the NNT was 50 in 2019, prior to the pandemic, broadly consistent with results from prevalence surveys (see e.g. ref [21]).

- *Amongst all individuals, regardless of symptom status, being tested in active case-finding.* In some settings, microbiological testing in active case-finding may extend beyond those with symptoms (that is, where active case-finding seeks to identify 'subclinical' as well as symptomatic TB). We modelled a scenario where the NNT is monitored amongst all individuals being tested, regardless of symptom status. For simplicity, we assumed that the NNT was the inverse of model-estimated prevalence throughout the period of simulation.

## Results

Fig A in S1 Text in the supporting information shows results of model calibration to pre-COVID data for each of the country groups, while Fig B in S1 Text shows the results for matching to quarterly notifications during COVID disruptions. As illustrated by all of these figures, the model is in reasonable agreement with the available data.

Figs 2–4 show the impact of post-COVID interventions in each country group, with the levels of coverage shown in Table 3. They illustrate the need for a combination of interventions acting at all stages of the TB cascade, including population-level prevention, in order to meet the 2030 goals for TB. Notably, the levels of vaccination coverage needed to meet the goals by 2030 varies by country group, being lower in the Sub-Saharan Africa country group (40%) than in others (72%). Fig C in S1 Text shows an alternative scenario where the vaccine is deployed only in 2030, five years later than shown in Fig 2, but with the same levels of coverage and pace of rollout. Concentrating on the example of the South Asia country group, this figure highlights how delayed deployment of the vaccine would also lead to delayed attainment of the SDG goals. In this country group, results suggest that the 2030 targets for incidence and mortality would only be met in 2033.

We next examined how progress towards the 2030 goals could be measured. First, additional analysis in the supporting information shows how, in the wake of COVID disruptions, the 2018 UN HLM targets for case detection may no longer be a reliable measure of progress towards the 2030 goals (see supporting information, section 5). In particular, in the South Asia country group, increased notifications in the coming years may arise from post-COVID increases in TB incidence, rather than from interventions. Conversely in the Sub-Saharan Africa country group, interventions that reduce incidence sufficiently rapidly can, in doing so, contribute towards reductions in notifications (Fig E in S1 Text). New targets will be required, and are in the process of development for the upcoming UN High-level meeting in September 2023 (Stop TB Partnership, personal communication).

To complement these efforts, we explored the alternative approach described above, of monitoring the number-needed to test (NNT) to identify 1 individual with TB. Figs 5–7 show results for each of the country groups. In general, the NNT grows with decreasing burden. An exception is NNT in routine programmatic settings in the Sub-Saharan Africa country group (Fig 6), where the NNT curve for improving diagnostics crosses the others shown. That is, even though each successive intervention scenario has increasing impact (as it represents a growing set of combined interventions), the NNT does not necessarily increase in a uniform way across the intervention scenarios. The reason is that in routine TB services, NNT can be

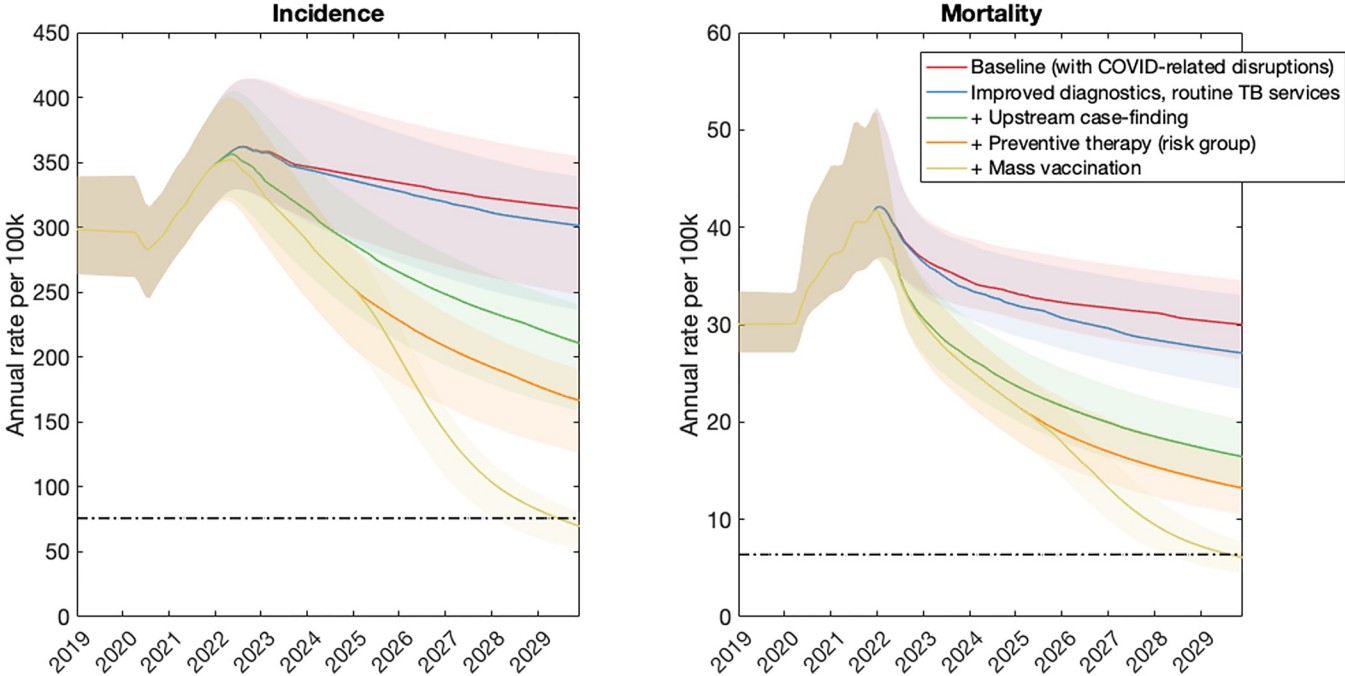

**Fig 2. Illustrative combination of interventions to meet the 2030 milestones, in the South Asia country group.** Interventions are shown in successive combination so that the bottom curve in each panel, for example, shows all interventions (including vaccination) acting together. As described in the main text, we assumed that vaccination is implemented from 2025 onwards, and that all other interventions are implemented from 2022 onwards. The horizontal, dashed line shows the target for incidence reduction by 2030. Under the intervention scenarios shown here, vaccination coverage would need to ensure that 72% of people with TB infection have vaccine-induced immunity, in order to meet the 2030 target.

affected by different factors in addition to TB burden, that complicate its use as a monitoring tool. For example, the use of new, more-sensitive diagnostic tools would detect greater numbers of TB amongst those being tested. While such beneficial effects would contribute to incidence declines, they would also cause a short-term drop in NNT, as a result of more individuals being diagnosed with TB amongst the existing influx of patients being tested. Thus, they would give rise to erroneous signals of growing TB burden. On the other hand, NNTs measured from active case-finding (middle and right panels, Figs 5–7) do not cross each other over time: assuming that ACF is conducted with the most accurate possible diagnostic tools, and that these tools are not changed over time, NNTs measured through ACF are therefore likely to be a more accurate representation of true trends in TB burden in the community, than NNTs measured in routine programmatic settings.

## Discussion

Our analysis highlights the range and intensity of interventions that will need to be employed, to meet the 2030 SDG goals. For example, modernising and coordinating routine TB services throughout the healthcare system (PPM; improved diagnostics; increased upfront DST) will form a critical foundation, improving patient outcomes and decreasing diagnostic delays. However, such interventions can only benefit patients who have already engaged with the healthcare system for their symptoms. Taking TB services proactively to those who have not yet had the opportunity to seek care (upstream case-finding) can bring about important reductions in TB incidence and mortality. However, even such efforts do not address the sizeable burden of latent TB infection that will continue to generate incident TB. Preventive therapy amongst eligible groups can bring about meaningful reductions in incidence and mortality,

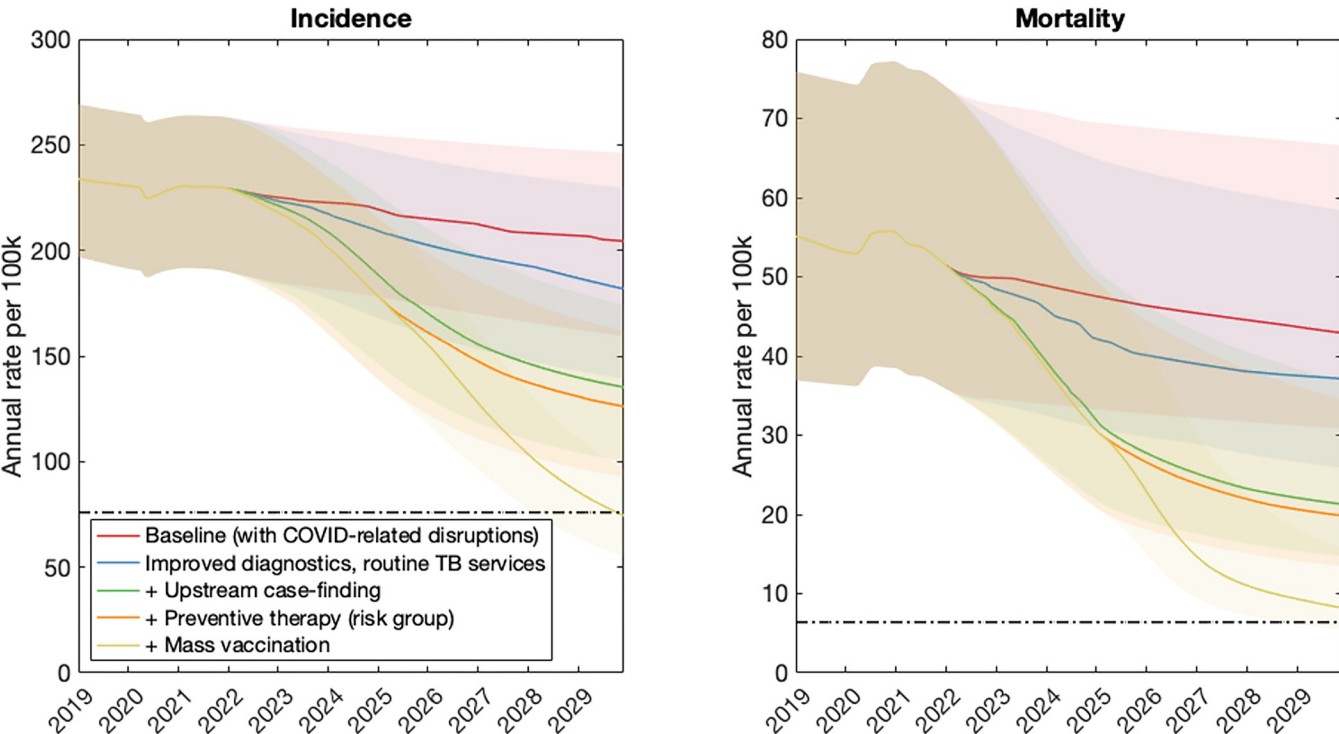

**Fig 3. Illustrative combination of interventions to meet the 2030 milestones, in the Sub-Saharan Africa country group.** Details are as in Fig 2. Under the intervention scenarios shown here, vaccination coverage would need to ensure that 40% of people with TB infection have vaccine-induced immunity, in order to meet the 2030 target.

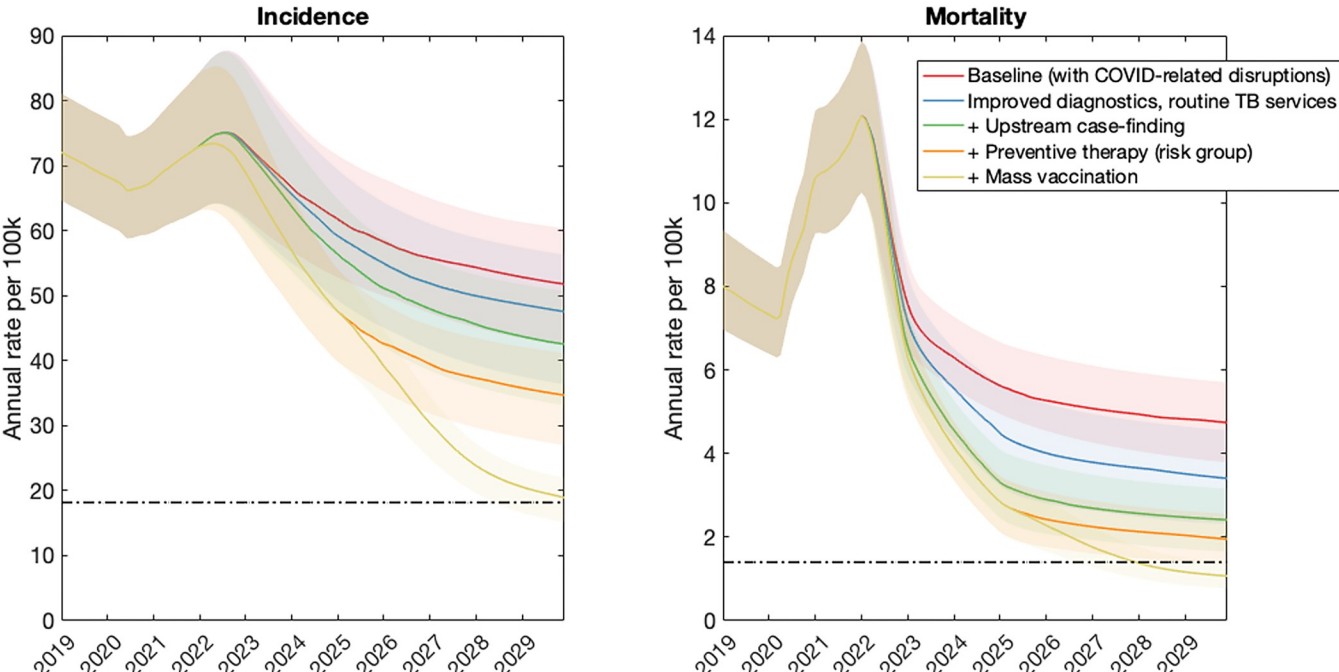

**Fig 4. Illustrative combination of interventions to meet the 2030 milestones, in the CAR/EU country group.** Details are as in Fig 2. Under the intervention scenarios shown here, vaccination coverage would need to ensure that 72% of people with TB infection have vaccine-induced immunity, in order to meet the 2030 target.

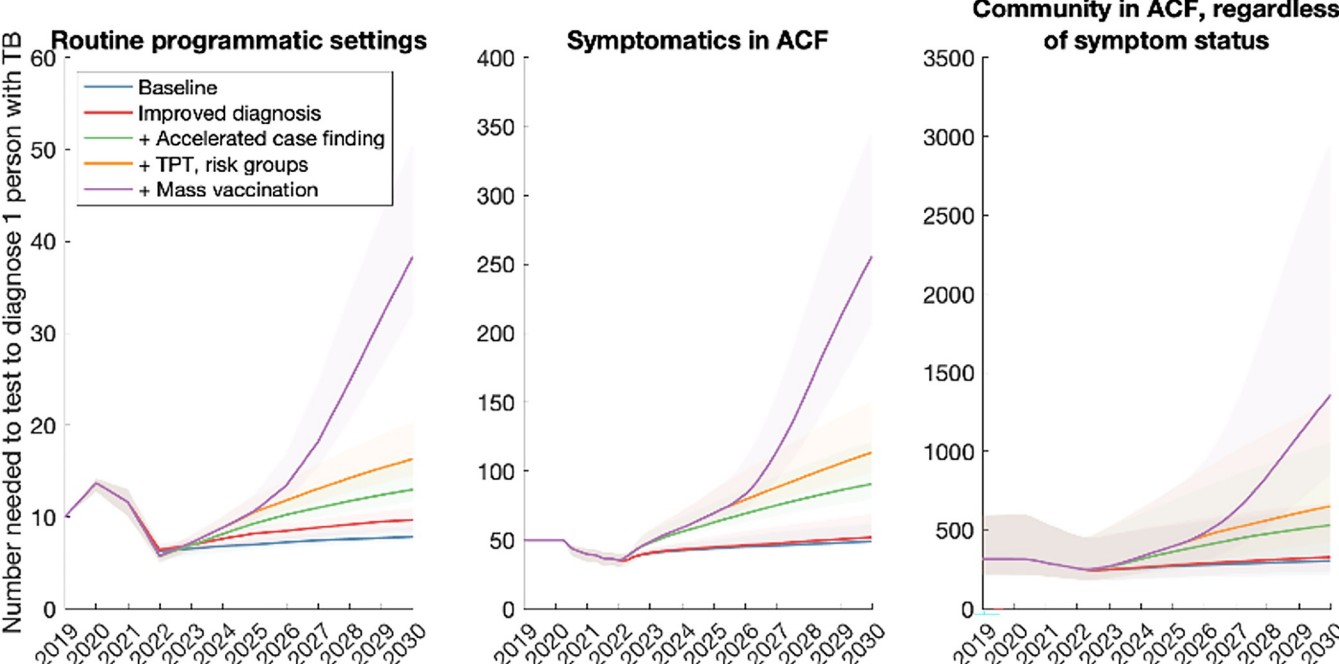

**Fig 5. Projected trends when tracking the number-needed-to-test (NNT) to identify 1 person with TB, in the South Asia country group.** Shown are three different examples of NNT: when monitoring patients being tested in routine TB care settings (left-hand panel); amongst those with symptoms who are screened as part of active case-finding (middle panel); and amongst all persons tested in active case-finding where eligibility is not restricted to those with symptoms (right-hand panel). As in Figs 2–4, all interventions are shown in cumulative combination, e.g. with the purple curve showing all interventions acting together.

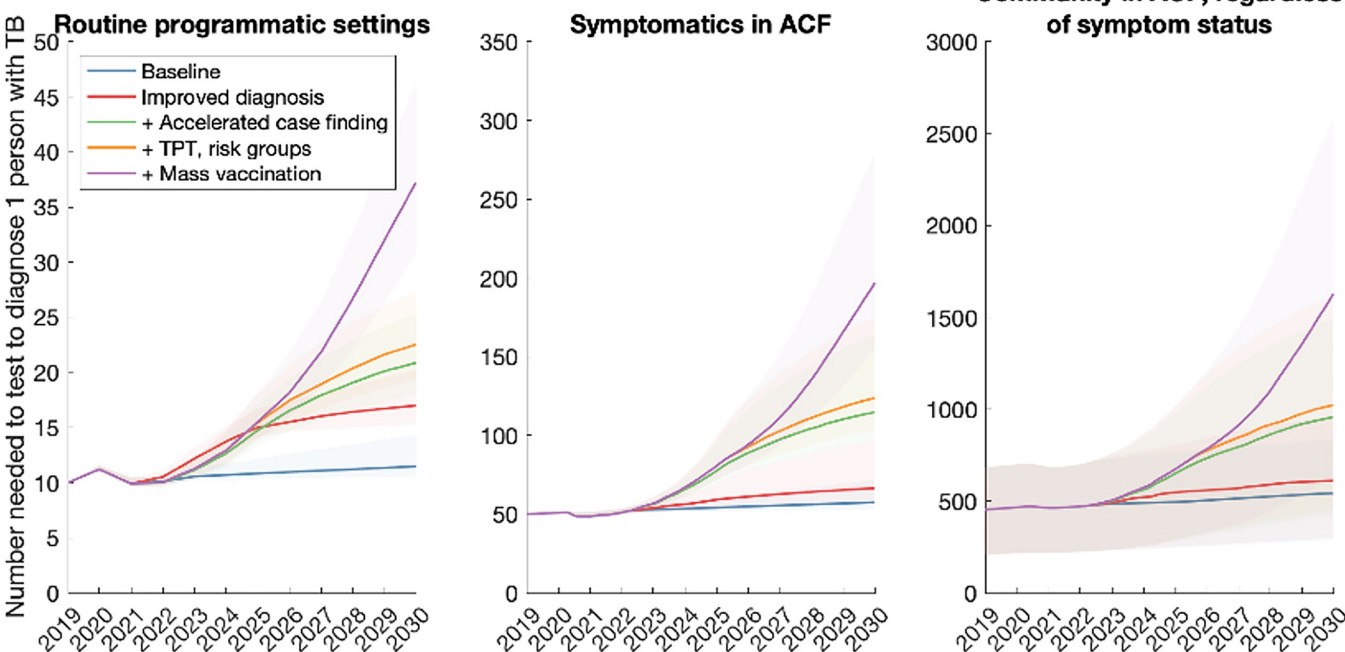

**Fig 6. Projected trends when tracking the number-needed-to-test (NNT) to identify 1 person with TB, in the Sub-Saharan Africa country group.** Details as in caption for Fig 5.

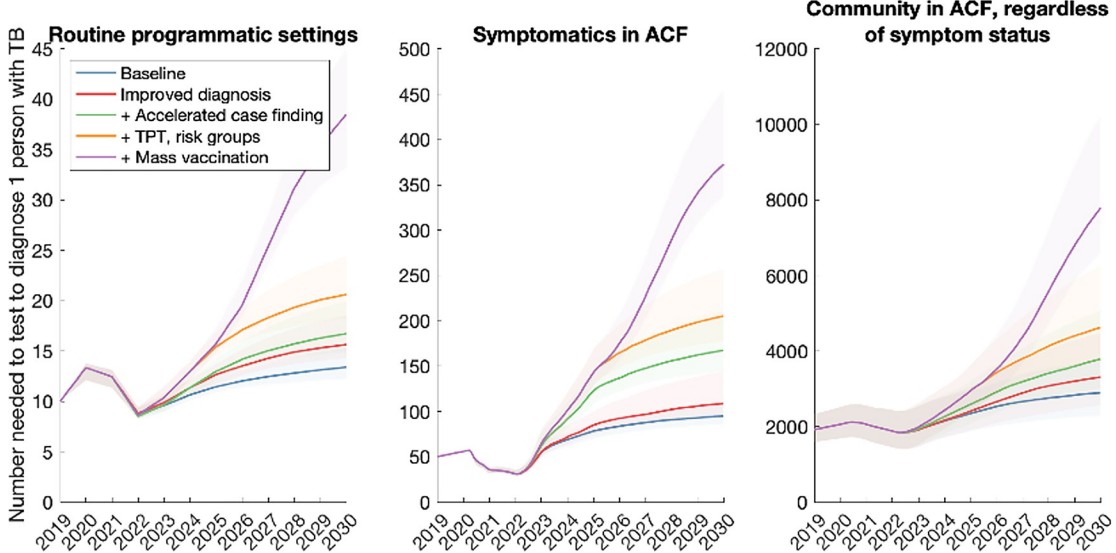

**Fig 7. Projected trends when tracking the number-needed-to-test (NNT) to identify 1 person with TB, in the Central Asia/ Europe country group.** Details as in caption for Fig 5.

but ultimately there will be a need to prevent TB in the general population, here modelled assuming the deployment of a post-exposure vaccine.

In order to meet these goals, our analysis also suggests that the 2018 UN HLM targets will need to be updated, in the wake of the COVID-19 pandemic (see supporting information, section 4). There will be substantial programmatic value in updated targets based on notifications, as these are already measured in routine TB services. However, our analysis illustrates how the number-needed-to-test to identify 1 patient with TB could provide a helpful complement to case-finding targets (Figs 5–7). Collecting such data need not involve substantial additional programmatic effort: Figs 2–4 shows the important role that could be played, by measures such as active case-finding (ACF, as a component of 'upstream case-finding'). Alongside the implementation of ACF, it will be straightforward to collect data on NNT to identify one patient with TB. Although NNTs could also be collected in facility-based settings, they may be subject to short-term variations that complicate their interpretation as a progress indicator (Fig 6).

Notably, although mass prevention (here modelled through vaccination) is necessary to meet the End TB goals in all country groups (Figs 2–4), the required coverage is lower in sub-Saharan Africa than in other country groups (Fig 3). A key reason for this result is the role of existing preventive measures amongst PLHIV; in a country group where 27% of TB incidence has HIV coinfection, ensuring all PLHIV receive TB preventive therapy will achieve a higher coverage of this preventive measure at the population level, than in other country groups. Thus, the incremental amount of vaccination required would be less than in other country groups. While not explicitly modelled here, similar impacts to vaccination might be achieved through other approaches for population-level prevention, such as addressing malnutrition and other societal determinants of TB, as well as widened use of shortened, simplified and safer preventive therapy.

As with any modelling study, our analysis has some limitations to note. Our models involve several simplifications: for example, ignoring age structure, and the difference between pulmonary and extrapulmonary TB. For simplicity, we have modelled countries at the level of

geographical aggregates. For national policy guidance, there will be a need for country-specific modelling, perhaps also addressing different priorities at the subnational level. One important limitation in our analysis is that it does not address the role of migration, or cross-border population movements more generally, in maintaining international TB transmission. Even if such factors are most important in low-TB burden settings [22, 23], they are likely to become increasingly relevant for high-burden countries that achieve substantial reductions in TB incidence and mortality substantially in the coming years. Incorporating these factors into modelling is an important area for future work. Another limitation is that certain countries do not fit neatly within the categories we have described, for example with Thailand having a high burden of HIV as having a large private sector. Worth reiterating is that these are illustrative, not definitive, scenarios: for example, if a country is capable of achieving a greater degree of 'upstream case-finding' than modelled here, its threshold vaccination coverage in order to meet the 2030 milestones would be lower than depicted in Figs 2–4. Finally, with a focus on epidemiological impact, our analysis does not address costs: given that the interventions considered here will be resource-intensive, it will be critical to ensure that they are implemented in the most cost-efficient way possible. Indeed, global cost estimates have recently been published in the Global Plan to End TB 2023–2030 [20], calling for mobilisation of USD 250 billion to reach the 2030 End TB goals globally. Importantly, the Global Plan also calls for an investment of at least USD 5bn per year to accelerate the development of new TB diagnostics, drugs and vaccines, new tools that will be critical for reaching the 2030 goals.

## Conclusion

Overall, this analysis illustrates how COVID-19 has hindered the global TB response: not only by leading to increases in global TB burden, but also complicating the monitoring of the progress towards TB elimination. Nonetheless, sharply accelerated and sustained scale-up in the global TB response would achieve substantial reductions in TB burden in the coming years. Such efforts could also open opportunities for new approaches for monitoring progress towards the End TB goals.

## Supporting information

**S1 Text. Additional technical details, figures and tables.**
(DOCX)

## Author Contributions

**Conceptualization:** Ya Diul Mukadi, Amy Bloom, Cheri Vincent, Sevim Ahmedov.

**Formal analysis:** Nimalan Arinaminpathy.

**Investigation:** Nimalan Arinaminpathy, Sevim Ahmedov.

**Methodology:** Nimalan Arinaminpathy.

**Validation:** Ya Diul Mukadi, Amy Bloom, Cheri Vincent, Sevim Ahmedov.

**Writing – original draft:** Nimalan Arinaminpathy, Sevim Ahmedov.

**Writing – review & editing:** Nimalan Arinaminpathy, Ya Diul Mukadi, Amy Bloom, Cheri Vincent, Sevim Ahmedov.

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
