## [Decision Letter · Decision Letter 0]

28 Apr 2023

PGPH-D-22-01675

Meeting the 2030 End TB goals in the wake of COVID-19: a modelling study of countries in the USAID TB portfolio

Dear Dr. Arinaminpathy,

Thank you for submitting your manuscript to PLOS Global Public Health. After careful consideration, we feel that it has merit but does not fully meet PLOS Global Public Health’s publication criteria as it currently stands. Therefore, we invite you to submit a revised version of the manuscript that addresses the points raised during the review process.

We look forward to receiving your revised manuscript.

Kind regards,

Hsiang-Yu Yuan, Ph.D.

Academic Editor

Journal Requirements:

2. Please send a completed 'Competing Interests' statement, including any COIs declared by your co-authors. If you have no competing interests to declare, please state "The authors have declared that no competing interests exist". Otherwise please declare all competing interests beginning with the statement "I have read the journal's policy and the authors of this manuscript have the following competing interests:"

3. We ask that a manuscript source file is provided at Revision. Please upload your manuscript file as a .doc, .docx, .rtf or .tex.

4. We have noticed that you have uploaded Supporting Information files, but you have not included a list of legends. Please add a full list of legends for your Supporting Information files after the references list.

Additional Editor Comments (if provided):

The study is very important for global public health. Please consider the suggestions and address the concerns given by both reviewers. 

Furthermore, Pre-COVID calibration data was used in this study. I am wondering, should you consider the modelling fitting step for the data as 'model training' or 'model calibration'?

Reviewers' comments:

Reviewer's Responses to Questions

**Comments to the Author**

1. Does this manuscript meet PLOS Global Public Health’s publication criteria? Is the manuscript technically sound, and do the data support the conclusions? The manuscript must describe methodologically and ethically rigorous research with conclusions that are appropriately drawn based on the data presented.

Reviewer #1: Partly

Reviewer #2: Yes

2. Has the statistical analysis been performed appropriately and rigorously?

Reviewer #1: Yes

Reviewer #2: N/A

3. Have the authors made all data underlying the findings in their manuscript fully available (please refer to the Data Availability Statement at the start of the manuscript PDF file)?

Reviewer #1: Yes

Reviewer #2: Yes

4. Is the manuscript presented in an intelligible fashion and written in standard English?

Reviewer #1: Yes

Reviewer #2: Yes

5. Review Comments to the Author

Reviewer #1: The authors applied mathematical modelling to answer how the countries in the USAID meet the End TB goals by 2030. The models consider different combination of interventions for different groups of countries. The authors applied the number-needed-to-test to identify one person with TB to provide a measure of the progress. The aim of the study and the conclusion are clear. However, the authors did not explain the modeling parts clearly.

For the modeling parts, there are some comments for the paper:

1. Page 2, introduction, the authors should provide a short discussion about the general picture of TB modelings.

2. Page 3, when the authors discussed the models, they should provide part of the models in the main text and refer to the full models in the supporting information.

3. In the supporting information, the explanation of the models is not complete. It does not show the uniqueness of the models they applied.

4. Page 3, “For simplicity we did not aim to model each country individually, and instead modelled interventions at the level of the country groups shown in table 1.” The authors consider three country groups so did the three groups interact with others? Also, within each country group, are the populations in different countries mixed together to be considered in the model? It seems that it does not satisfy the real situation?

5. Do the ranges of the values in Table 2 come from the real data or depend on some assumption?

6. Are there any reference to decide the ranges of the model parameters? For example, some of them were set +/- 10%, and some were +/-25%.

Reviewer #2: This paper sought to estimate the coverage of TB-specific interventions that would be required to meet the 2030 End TB Goals. It also sought to evaluate the reliability of potential new indicators for assessing progress, beyond the usual TB notification indicator.

This is a really interesting manuscript and would be an important and useful addition to the literature. I think it should be published and I have made, minor suggestions below, with the primary aim of improving its clarity and utility.

Disclosures

The submitting author states ‘This work was supported by USAID. YDM, AB, CV and SA are employees of USAID; the funders otherwise had no role in the study’. Given they are all authors on the manuscript, and I am sure they contribute more than enough to take on that role, this disclosure must be untrue.

Suggest modifying to state the role in funding and authorship more clearly

Line 113 ‘We assumed all interventions would be rolled out starting in 2022’

Not true. Vx later. Needs correcting

Line 118.

Unlike the rest of the manuscript, this is a rather wordy and unclear sentence. Could it be made more clear?

Line 161. Starting ‘In each fig…

Delete sentence. Not needed. These #s are quoted in the subsequent text

Line 167

It would be better (IMHO) to move the explanatory text on why lower vx coverage is required in SSA from the discussion into the results. As it is based on an analysis of model results, it could fairly be defined as a result, and that is where the reader will have the question

Line 168 - required coverage if 2030 vx roll out assumed

It would be useful to add the coverages assumed (100%?) and year the 2030 targets met, into the main text, for all regions. This is an important point given the long delays to the start of the M72 Ph3 trial

Line 176 - notifications no longer reliable

It needs a couple more sentences here in the main text to say why notifications no longer reliable (not just in supporting material)

Line 238 - would be sensible to add something like ‘or more widespread use of safer preventive therapy’, to this sentence

Line 255

Would be great to add something like ‘, and mobilsing the $5bn/ year that is also needed for R&D to get the new vaccine etc, else targets certainly won't be met’

Line 274. Table. For row ‘TB preventive therapy, risk groups’,

need to add the assumed efficacy to the description box

Line 274. Table. For row ‘TB vaccine’,

need to add the assumed duration of efficacy, & that coverage was calculated to meet End TB targets, to the description box

6. PLOS authors have the option to publish the peer review history of their article (what does this mean?). If published, this will include your full peer review and any attached files.

**Do you want your identity to be public for this peer review?** For information about this choice, including consent withdrawal, please see our Privacy Policy.

Reviewer #1: No

Reviewer #2: **Yes: **Richard White

---

## [Decision Letter · Decision Letter 1]

14 Aug 2023

Meeting the 2030 End TB goals in the wake of COVID-19: a modelling study of countries in the USAID TB portfolio

PGPH-D-22-01675R1

Dear Dr Arinaminpathy,

We are pleased to inform you that your manuscript 'Meeting the 2030 End TB goals in the wake of COVID-19: a modelling study of countries in the USAID TB portfolio' has been provisionally accepted for publication in PLOS Global Public Health.

Best regards,

Hsiang-Yu Yuan, Ph.D.

Academic Editor

Both reviewers agreed this work to be published. The article is significant. Congratulations on the acceptance of your paper.

Reviewer Comments (if any, and for reference):

Reviewer's Responses to Questions

**Comments to the Author**

1. If the authors have adequately addressed your comments raised in a previous round of review and you feel that this manuscript is now acceptable for publication, you may indicate that here to bypass the “Comments to the Author” section, enter your conflict of interest statement in the “Confidential to Editor” section, and submit your "Accept" recommendation.

Reviewer #1: All comments have been addressed

Reviewer #2: All comments have been addressed

2. Does this manuscript meet PLOS Global Public Health’s publication criteria? Is the manuscript technically sound, and do the data support the conclusions? The manuscript must describe methodologically and ethically rigorous research with conclusions that are appropriately drawn based on the data presented.

Reviewer #1: Yes

Reviewer #2: Yes

3. Has the statistical analysis been performed appropriately and rigorously?

Reviewer #1: N/A

Reviewer #2: Yes

4. Have the authors made all data underlying the findings in their manuscript fully available (please refer to the Data Availability Statement at the start of the manuscript PDF file)?

Reviewer #1: Yes

Reviewer #2: Yes

5. Is the manuscript presented in an intelligible fashion and written in standard English?

Reviewer #1: Yes

Reviewer #2: Yes

6. Review Comments to the Author

Reviewer #1: All the comments have been addressed.

Reviewer #2: Authors have addressed my comments successfully.

7. PLOS authors have the option to publish the peer review history of their article (what does this mean?). If published, this will include your full peer review and any attached files.

**Do you want your identity to be public for this peer review?** For information about this choice, including consent withdrawal, please see our Privacy Policy.

Reviewer #1: No

Reviewer #2: **Yes: **Richard White
